# Dependence of Body Stability on Optical Conditions during VR Viewing

Gi-Seong Jeong [1], Hyun-Goo Kang [2,*,†] and Sang-Yeob Kim [3,*,†]

[1] Department of Medical Health Science, Graduate School, Kangwon National University, Samcheok 25949, Republic of Korea; optique_signature@naver.com
[2] Department of Optometry, Catholic Kwandong University, Gangneung 25601, Republic of Korea
[3] Department of Optometry, College of Health Science, Kangwon National University, Samcheok 25949, Republic of Korea
[*] Correspondence: hgkang@cku.ac.kr (H.-G.K.); syk@kangwon.ac.kr (S.-Y.K.); Tel.: +82-33-649-7375 (H.-G.K.); +82-33-540-3413 (S.-Y.K.)
[†] These authors contributed equally to this work.

**Abstract:** The dependence of body stability on the distance between the optical centers of VR-device lenses and the refractive error status of users during VR viewing was investigated. Participants included 31 adults, and their postural-control ability was measured using a BTrackS device. The optical conditions were (1) COCD (comfortable optical center distance), (2) COCD+2D (comfortable optical center distance with 2D myopia), (3) COCD-2D (comfortable optical center distance with 2D hyperopia), (4) DOCD (uncomfortable optical center distance), (5) DOCD+2D (uncomfortable optical center distance with 2D myopia), and (6) DOCD-2D (uncomfortable optical center distance with 2D hyperopia). Posture was assessed under these six optical conditions while the participants were wearing a VR device and watching a 3D roller-coaster video. The sway-path length was significantly increased under the COCD-2D, DOCD, DOCD+2D, and DOCD-2D conditions compared to the COCD condition ($p < 0.05$). In the case of maximum sway velocity, the results showed significant increases under the DOCD, DOCD+2D, and DOCD-2D conditions compared to the COCD condition ($p < 0.05$). The analysis revealed that when users are viewing VR displays, optimization of the distance to the optical center of the VR-device lenses and correction of the refractive errors for individual users was a significant factor in minimizing body instability.

**Keywords:** virtual reality; body stability; sway-path length; sway velocity; refractive errors; distance between VR-device lenses

## 1. Introduction

Virtual reality (VR) is a technology that allows users to experience interactions among sight, hearing, body movement, and touch by using computer graphics to render a simulated three-dimensional space that closely resembles the real world [1]. As a result of recent developments, the public can experience VR using cardboard-type VR devices that are able to create VR using smartphones. The head-mounted display (HMD) of the VR devices is composed of a smartphone display and optical lenses with +17 D~+20 D. The HMD method creates minute binocular disparity by generating two separate video screens and three-dimensional depth by fusing the images that reach the retina of each eye [2].

VR is widely applied in education, medicine, and industry because it offers the distinct advantage of being able to create an environment similar to the real environment, thus providing users with indirect experiences. However, the most formidable barrier the VR industry will have to overcome for qualitative development is the problem of cyber sickness. Sensory-conflict theory proposes that the principal factor responsible for inducing cyber sickness is conflict among the different types of sensory information, such as visual, vestibular, and proprioceptive cues in the brain [3]. According to this theory,

cyber sickness arises as a result of conflicts between visual and vestibular cues because the experience perceived in the virtual environment is more artificial than that in the real world. Song [4] proposed a method to mitigate cyber sickness that involved analyzing the relationship between cyber sickness and reflective eye movements. Chang [5] conducted research from an integrated perspective by linking the causes of cyber sickness, the human cognitive system, and methods of measuring cyber sickness. However, ongoing research on methods to alleviate cyber sickness is necessary because the causes thereof have not yet been fully elucidated.

As we usher in the digital age, it is necessary to develop human-friendly 3D virtual environments. As such, minimizing cyber sickness has become a critical task in the field of optometry. Hong et al. [6] conducted eye-movement and fusion-ability tests using a VR device and analyzed the correlation with conventional visual-function tests. They concluded that VR devices can be a useful alternative to traditional methods of visual-function assessment. Cho et al. [7] investigated the relationship between cyber sickness and visual function and analyzed whether sensory training can reduce these symptoms. Their results showed that the stereopsis, ocular alignment, accommodative function, and vergence function were significantly lower in the group with severe cyber-sickness symptoms. Additionally, the sensory-training program was proven to have a positive effect, alleviating cyber sickness.

Another theory suggests that postural instability, which continuously occurs in VR environments, can also trigger cyber sickness [3]. Humans inherently adjust their posture according to the environment, but in unfamiliar environments such as virtual reality, they may not be able to effectively utilize their existing posture-control strategies. Our understanding of how optical conditions while watching VR images affect postural-control ability is still insufficient. This knowledge gap prompted our study, in which we attempt to analyze the effect of optical conditions such as the distance to the optical centers of VR-device lenses and the refractive-error-correction state on body stability while watching VR images and to provide useful information to VR-device manufacturers and users.

## 2. Materials and Methods

### 2.1. Subjects

Thirty-one young adults (13 males and 18 females) with an average age of 22.73 years participated in this study. All participants had monocular best-corrected visual acuity of 0.9 or higher and did not have any neurological, neuromuscular, musculoskeletal, or systemic diseases related to body balance. Additionally, the study included only adults who did not report any subjective symptoms related to binocular vision dysfunction or accommodative dysfunction and had no history of eye diseases or of the use of related medication. All participants received appropriate verbal and written explanations of the purpose and methods of the study, and their consent was obtained prior to the conduction of the experiment. The flow chart of methods used in the present study is shown in Figure 1.

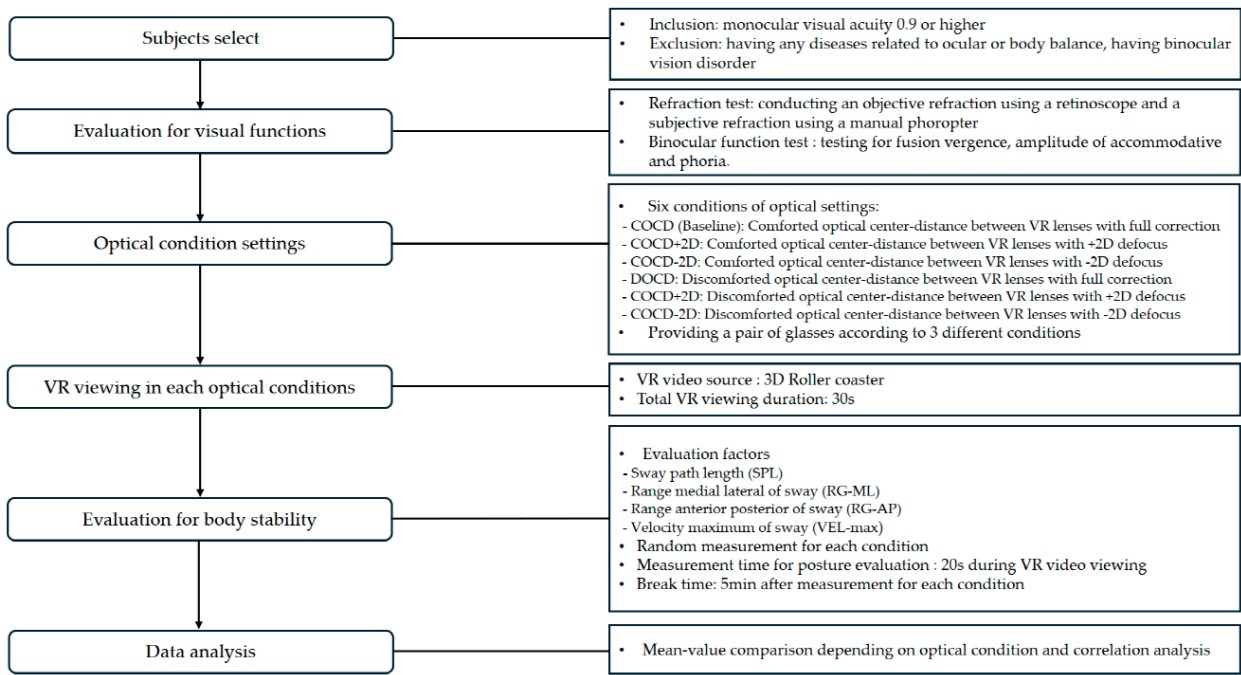

**Figure 1.** The flow chart of the present study.

*2.2. Experimental Devices*

In this experiment, an HMD VR device (VR BOSS Xtrek, Neofeel, Yongin, Republic of Korea and a smartphone (LM-G710N, LG Electronics, Seoul, Republic of Korea) were used. The diagonal of the smartphone display was 154.7 mm long, and the resolution of the display was 3120 × 1440 pixels with 564 ppi. The VR device could support a maximum smartphone size of 195 mm × 88 mm. The distance between the eyes and the lenses could be controlled. The distance to the optical centers could also be controlled in the range 60–70 mm. The lenses on the VR device were aspherical, had +17 D spherical refraction, and provided a field of view of approximately 100°.

A BTrackS balance plate (Balance Tracking Systems, Inc., San Diego, CA, USA) was used to evaluate the dependence of the body stability on the optical conditions while watching VR. The BTrackS balance plate, a device used to measure body stability, offers reliability and accuracy of more than 99.9% compared with other devices [8]. The body center of pressure (COP) represents the body's sway path by tracking changes in the center of pressure as the subject stands on the plate to measure the body stability [9]. Movement patterns such as the length, speed, and area were comprehensively analyzed by converting the sway-path length into centimeters and displayed using the associated software [9].

*2.3. Evaluation Factors for Visual Functions*

The fully corrected refraction of each subject was determined by conducting an objective refraction test using an Elite retinoscope, Welch Allyn, USA, and a subjective refraction test at a distance of 5 m using a manual phoropter (Essillor MPH-150E, Essilor Instruments, Charenton-le-pont, France) and an LCD polar (24 inch) chart. The subjective refraction test was conducted using the fogging method with a 0.5 target visible after the maximum spherical refraction test. Astigmatism was tested using a radiographic target in myopic double astigmatism. Precision astigmatism was assessed using a Jackson cross cylinder at the spherical refraction where the maximum plus to maximum visual acuity (MPMVA) was obtained. Binocular accommodative balance was tested using the fogging method. The final fully corrected refraction was determined based on the MPMVA method [10].

Distant heterophoria was tested using the passive phoropter, the LCD polar chart at a distance of 5 m, and a modified Torrington target. The fusion vergence test, one of the methods used to evaluate the ability to maintain binocular single vision, is a method used to

quantitatively evaluate the range of convergence power of eyes that can maintain binocular single vision [11]. Each positive fusional vergence (PFV, convergence) and negative fusional vergence (NFV, divergence) was tested once at a distance of 5 m using the passive phoropter and the LCD polar chart. The total accommodative amplitude of each subject was tested using a push-up method. The distance to the point at which the target appeared blurry for the first time was measured and converted to diopters by allowing each subject, while wearing a fully corrected pair of glasses, to see the target clearly and then moving the target [12].

### 2.4. Evaluation Factors for Body Stability

#### 2.4.1. Sway-Path length (SPL)

The sway-path length is the sum of 500 values sampled for 25 s at 25 Hz, and each value is the inverse of the sum of the square of the difference between COPx2 and COPx1 (average position of the left and right COPs relative to the center) and the square of the difference between COPy2 and COPy1 (average position of front and back COPs relative to the center) [13]. Specifically, the sway-path length is the sum of all deviations from the center of pressure (expressed in cm) when the body sways. This value could replace the magnitude of body sway, and the greater the sway-path length, the greater the body sway.

#### 2.4.2. Range Medial Lateral of Sway (RG-ML)

Tracking the COPs and expressing them in a single image yields the range of the medial lateral of sway, the inner and outer sway ranges in the image, which is the calculated value of the distance expressed in cm between the maximum position and the minimum position of the signals measured along a particular axis [14]. The range of the medial lateral of sway is the range that moved most to the right or left among the measured body-sway paths. It is always expressed in positive numbers because it is the right or left range among the measured sway paths.

#### 2.4.3. Range Anterior Posterior of Sway (RG-AP)

Tracking the COPs and expressing them in a single image yields the range of the medial anterior posterior of sway, the upper and lower sway ranges in the image, which is the calculated value of the distance expressed in cm between the maximum and minimum positions of the signals measured along a particular axis. The range medial lateral of sway is the range that moved most to the anterior or posterior direction among the measured body sway paths. It is always expressed in positive numbers because it is the anterior or posterior range among the measured sway paths.

#### 2.4.4. Velocity Maximum of Sway (VEL-Max)

The velocity maximum is defined as the maximum value between two zero-crossing points. The anterior and values on the right are positive maxima, whereas the posterior and values on the left are negative maxima. High absolute values of the velocity maximum indicate difficulty maintaining body stability because of fast swaying. The anterior and velocity maxima to the right are expressed in positive (+) numbers, and the posterior and velocity maxima to the left are expressed in negative (−) numbers [15].

### 2.5. Experimental Procedures

First, the test chart designed in a previous study was used in this experiment in order to set the conditions related to the optical center distance (OCD) of the VR optical lenses (Figure 2). The test chart was produced by applying the Torrington chart [6]. The test subjects were asked to stare at the chart in the VR device through the smartphone and to manipulate the OCD adjustment dial in order to position the diagonal-line target on the right in the center of the cross target on the left. The most comfortable OCD condition was defined as the point where the cross and the diagonal lines were aligned, and the most uncomfortable OCD condition was defined as the point at which the separation between

the cross and the diagonal lines reached its maximum. The final OCD conditions were determined after confirming that the participants could clearly distinguish between the comfortable and uncomfortable conditions through subjective perception.

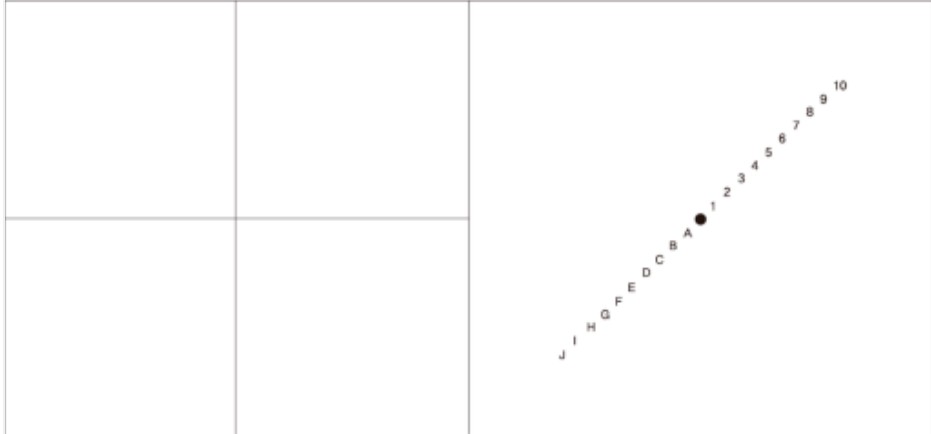

**Figure 2.** Test chart to determine the center distance of VR optical lenses [16].

In this study, the following six conditions were defined to take into account the combined effect of the optical center distance and refractive errors of the VR-device lenses.

(1) Comfortable optical center distance with full correction (COCD): The viewer feels most comfortable.
(2) Comfortable optical center distance with OU: S+2D defocus (COCD+2D): The viewer feels most comfortable in a 2D myopia-inducing condition.
(3) Comfortable optical center distance with OU: S-2D defocus: The viewer feels most comfortable in a 2D hyperopia-inducing condition.
(4) Uncomfortable optical center distance with full correction (DOCD): The viewer feels most uncomfortable with full correction.
(5) Uncomfortable optical center distance with OU: S+2D defocus (DOCD+2D): The viewer feels most uncomfortable in a 2D myopia-inducing condition.
(6) Uncomfortable optical center distance with OU: S-2D defocus (DOCD-2D): The viewer feels most uncomfortable in a 2D hyperopia-inducing condition.

Eyeglasses, fabricated to accommodate the error-causing refractive conditions, were provided to each test subject. All test subjects were asked to stand barefoot on the BTrackS balance plate in order to measure their body stability under the six optical conditions. For the posture assessment, the test subjects were asked to align the heels of their feet with the central line of the measurement plate with their hands akimbo. Afterwards, they were asked to wear a VR device under the six optical conditions and to watch a roller-coaster video in 3D mode. The roller-coaster video used this study was designed for 360-degree virtual reality (VR) in 4K resolution, 60 frames per second (fps), and a 3D format, which was presented to both eyes simultaneously. During the posture assessment, the same 30 s segment from the entire roller-coaster video was edited and provided for viewing to all participants. The subjects' body stability was measured for 20 s while the subjects were watching the video. The optical conditions were applied in random order when the measurements were conducted. A 10 min rest period was allowed after measurement under each condition.

*2.6. Data Analysis*

SPSS (Version 24 for window, SPSS Inc., Chicago, IL, USA) was used for the statistical analysis of data. Repeated-measures ANOVA was used for the comparative analysis of the subjects' body stability under each of the optical conditions. Pearson's correlation analysis was used to identify the trends in relationships between each of the visual functions and

the factors used to evaluate body stability. In all of the analyses, results were considered statistically significant when *p* < 0.05.

## 3. Results

### 3.1. Changes in the Sway-Path Length under Each of the Optical Conditions

Figure 3 shows the variations in the sway-path length depending on the six optical conditions defined in this study. The sway-path length increased significantly under the COCD-2D, DOCD, DOCD+2D, and DOCD-2D conditions compared to the COCD condition (repeated-measures ANOVA, F = 3.887/*p* = 0.009). A post-hoc analysis revealed that *p* = 0.012 for COCD vs. COCD-2D, *p* = 0.001 for COCD vs. DOCD, *p* = 0.015 for COCD vs. DOCD+2D, and *p* = 0.001 for COCD vs. DOCD-2D. However, while the sway-path length tended to increase compared to the COCD+2 condition, there was no statistically significant difference (*p* = 0.258 for COCD vs. COCD+2D).

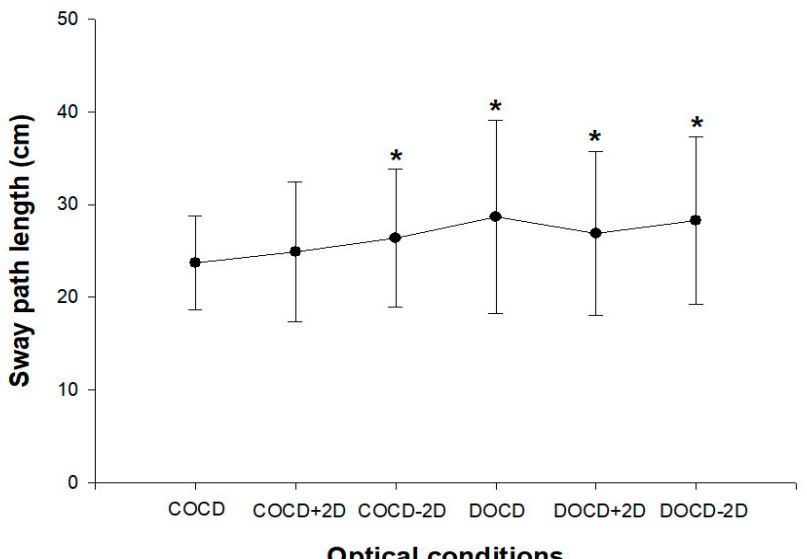

**Figure 3.** Changes in the sway-path length depending on the optical conditions during VR viewing. * *p* < 0.05: significantly different compared to COCD (baseline) by LSD post-hoc analysis of repeated = measures ANOVA. Error bars were expressed as mean ± SD. N = 31 (for each condition). COCD (baseline): comfortable optical center distance with full correction for refractive errors. COCD+2D: comfortable optical center distance with 2-diopter-induced myopia. COCD-2D: comfortable optical center distance with 2-diopter-induced hyperopia. DOCD: uncomfortable optical center distance with full correction. DOCD+2D: uncomfortable optical center distance with 2-diopter-induced myopia. DOCD-2D: uncomfortable optical center distance with 2-diopter-induced hyperopia.

### 3.2. Changes in the Range of Sway Lengths under Each of the Optical Conditions

Figure 4 shows the variations in the range of sway lengths under the six optical conditions. The variations in the range of anterior-posterior sway were significantly different only under the DOCD-2D condition compared to the COCD condition (repeated-measures ANOVA, F = 2.732/*p* = 0.032, Figure 4A). A post-hoc analysis revealed that *p* = 0.019 for COCD vs. COCD-2D. The variations in the range of medial-lateral sway were not statistically significant (repeated-measures ANOVA, F = 0.718/*p* = 0.616, Figure 4B).

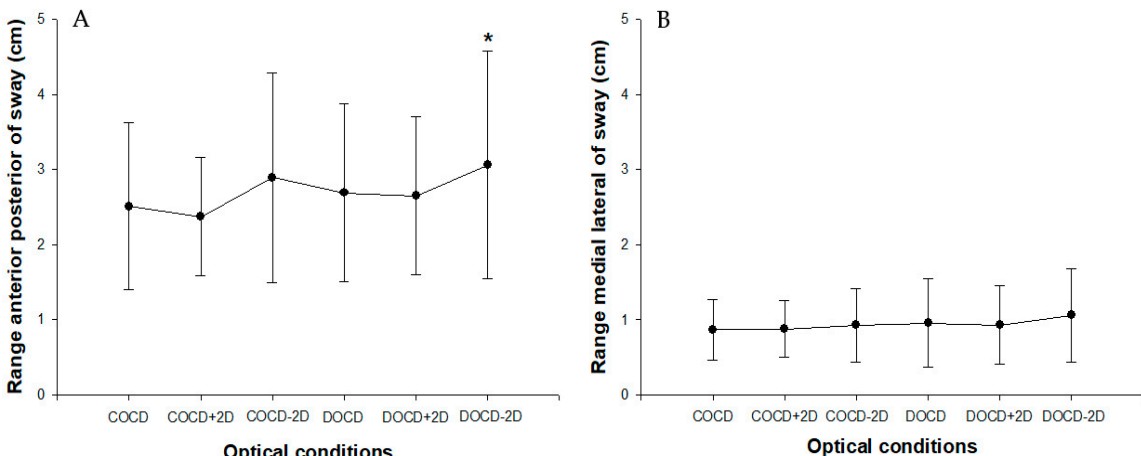

**Figure 4.** Changes in the sway range depending on the optical conditions during VR viewing. * $p < 0.05$: significantly different compared to COCD (baseline) by LSD post-hoc analysis of repeated-measures ANOVA. Error bars were expressed as mean $\pm$ SD. N = 31 (for each condition). (**A**): Anterior-posterior (**B**): Medial-lateral. COCD (base line): comfortable optical center distance with full correction for refractive errors. COCD+2D: comfortable optical center distance with 2-diopter-induced myopia. COCD-2D: comfortable optical center distance with 2-diopter-induced hyperopia. DOCD: uncomfortable optical center distance with full correction. DOCD+2D: uncomfortable optical center distance with 2-diopter-induced myopia. DOCD-2D: uncomfortable optical center distance with 2-diopter-induced hyperopia.

### 3.3. Changes in the Velocity Maximum of Sway under Each of the Optical Conditions

Figure 5 shows the variations in the velocity maximum of sway. The differences in the velocity maximum of sway were statistically significant under the conditions of DOCD, DOCD+2D, and DOCD-2D compared to that of COCD condition (repeated-measures ANOVA, F = 4.030/$p$ = 0.008). A post-hoc analysis revealed that $p$ = 0.001 for COCD vs. DOCD, $p$ = 0.007 for COCD vs. DOCD+2D, and $p$ = 0.005 for COCD vs. DOCD-2D condition.

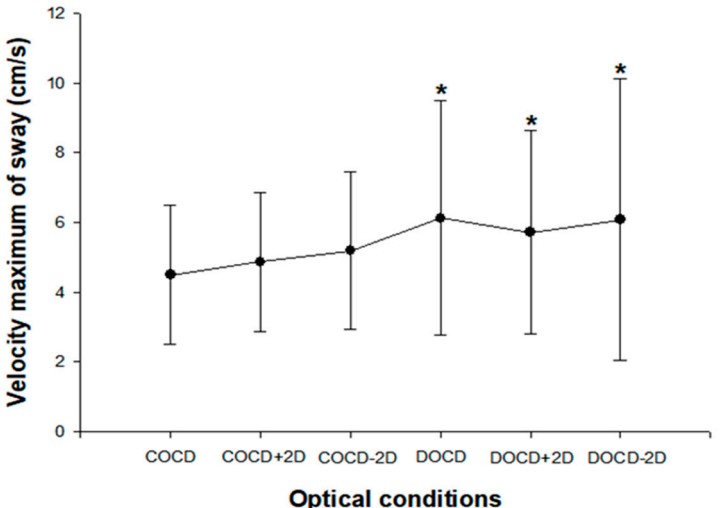

**Figure 5.** Changes in the sway velocity depending on the optical conditions during VR viewing. * $p < 0.05$: significantly different compared to COCD (baseline) by LSD post-hoc analysis of repeated

-measures ANOVA. Error bars were expressed as mean ± SD. N = 31 (for each condition). COCD (base line): comfortable optical center distance with full correction for refractive errors. COCD+2D: comfortable optical center distance with 2-diopter-induced myopia. COCD-2D: comfortable optical center distance with 2-diopter-induced hyperopia. DOCD: uncomfortable optical center distance with full correction. DOCD+2D: uncomfortable optical center distance with 2-diopter-induced myopia. DOCD-2D: uncomfortable optical center distance with 2-diopter-induced hyperopia.

*3.4. Analysis of the Correlation between Visual Functions and Body-Stability Evaluation Factors under Each of the Optical Conditions*

Tables 1 and 2 present the results of the analysis of correlation between visual functions and body-stability evaluation factors under each of the optical conditions. The results shown in Table 1 indicate that a weak negative correlation exists between the separation point of distant positive convergence and the velocity maximum of sway ($r$ = 0.364/ $p$ = 0.044). However, none of the visual functions was correlated to the body-stability evaluation factor under any of the conditions in the DOCD group (Table 2).

**Table 1.** Analysis of relationship between the factors responsible for body stability and visual functions in the COCD group during VR viewing.

| Optical Condition | Factor of Balance (Unit) | Visual Factors (Unit) | | | |
|---|---|---|---|---|---|
| | | Phoria (△) | AA (D) | BI Break (△) | BO Break (△) |
| COCD | Sway length (cm) | −0.277/0.132 | −0.133/0.546 | −0.154/0.407 | −0.154/0.407 |
| | DIS-max (cm) | 0.003/0.987 | −0.63/0.737 | 0.002/0.992 | −0.275/0.135 |
| | VEL-max (cm/s) | −0.057/0.761 | −0.86/0.645 | −0.084/0.652 | −0.153/0.412 |
| COCD+2 | Sway length (cm) | −0.211/0.254 | 0.014/0.940 | −0.103/0.582 | −0.103/0.582 |
| | DIS-max (cm) | −0.166/0.371 | 0.124/0.505 | 0.114/0.540 | −0.224/0.225 |
| | VEL-max (cm/s) | −0.187/0.313 | −0.003/0.985 | −0.054/0.774 | −0.049/0.792 |
| COCD-2 | Sway length (cm) | −0.087/0.641 | −0.058/0.758 | −0.134/0.474 | −0.218/0.239 |
| | DIS-max (cm) | −0.307/0.093 | −0.037/0.844 | −0.062/0.740 | −0.364 */0.044 |
| | VEL-max (cm/s) | 0.111/0.551 | −0.072/0.701 | −0.230/0.213 | −0.067/0.722 |

Data are expressed as $r$-value of Pearson's correlation coefficient/$p$-value. * $p$ < 0.05: significantly different from Pearson's correlation coefficient. N = 31 (for each condition). AA: accommodative amplitude.

**Table 2.** Analysis of relationship between the factors responsible for body stability and visual functions in the DOCD group during VR viewing.

| Optical Condition | Factor of Balance (Unit) | Visual Factors (Unit) | | | |
|---|---|---|---|---|---|
| | | Phoria (△) | AA (D) | BI Break (△) | BO Break (△) |
| DOCD | Sway length (cm) | −0.293/0.109 | −0.107/0.565 | −0.112/0.548 | −0.179/0.336 |
| | DIS-max (cm) | −0.089/0.635 | 0.040/0.833 | −0.113/0.544 | −0.172/0.354 |
| | VEL-max (cm/s) | −0.330/0.070 | −0.155/0.404 | −0.085/0.650 | −0.222/0.230 |
| DOCD+2 | Sway length (cm) | −0.114/0.541 | −0.122/0.515 | −0.021/0.911 | −0.097/0.602 |
| | DIS-max (cm) | 0.008/0.964 | −0.019/0.920 | 0.159/0.393 | 0.003/0.986 |
| | VEL-max (cm/s) | −0.047/0.801 | −0.190/0.305 | 0.056/0.766 | −0.072/0.701 |
| DOCD-2 | Sway length (cm) | 0.139/0.455 | −0.093/0.619 | −0.103/0.580 | −0.272/0.139 |
| | DIS-max (cm) | −0.094/0.617 | −0.004/0.983 | 0.088/0.636 | −0.234/0.204 |
| | VEL-max (cm/s) | 0.072/0.700 | −0.154/0.408 | −0.015/0.936 | −0.235/0.202 |

Data are expressed as $r$-value of Pearson's correlation coefficient/$p$-value. N = 31 (for each condition). AA: accommodative amplitude.

## 4. Discussion

In order to maintain body stability, the visual system provides specific information about the visual environment, such as the ground level, distance, depth, and spatial po-

sition [17]. The vestibular system, through the semicircular canals and gravity receptors, provides information about one's position and direction of movement [18]. The somatosensory system senses the movement and position of the body and controls the body's balance using external stimuli through peripheral sensory receptors such as tendons and muscles [19,20]. Cyber sickness is caused by a conflict between visual sensing and vestibular sensing because the experiences sensed in the VR environment are more artificial than those in the real environment. In an unfamiliar environment such as the VR environment, body stability is impeded because the existing posture-control strategies cannot be exercised smoothly [3]. In this study, we analyzed the effect of viewing VR images on the body stability depending on the optical conditions, specifically the optical center distance and refractive error correction in the VR device.

The results of the comparison and analysis of the average body-stability evaluation factors and the extent to which they depended on each optical condition defined in this study are shown in Figures 3–5. The main findings of these analyses are as follows. First, the sway-path length increased significantly under the COCD-2D, DOCD, DOCD+2D, and DOCD-2D conditions compared to the baseline condition, COCD ($p < 0.05$, Figure 3). The range of sway in the anterior and posterior directions increased only under the DOCD-2D condition compared to the DOCD condition ($p < 0.05$, Figure 4A). The maximum velocity of sway increased significantly under the DOCD, DOCD+2D, and DOCD-2D conditions when compared to the DOCD condition ($p < 0.05$, Figure 5). An analysis of all the results together revealed that significant differences were found in most measurement factors when the test subjects felt uncomfortable about the optical distance conditions or under hyperopia-inducing conditions. On the other hand, a peculiar finding of note was that the results of the myopia-inducing conditions were similar to those of COCD condition. Tables 1 and 2 present the results of our analyses to determine the extent to which the correlation between visual functions and body-stability evaluation factors depended on each optical condition. The visual functions included in this study were heterophoria, accommodation, and convergence. No correlations were found to exist between each visual function and body-sway evaluation factor for the COCD, COCD+2D, DOCD, DOCD+2D, and DOCD-2D conditions. A weak negative correlation was found between the separation point of far-distance positive convergence and the velocity maximum of sway ($r = 0.364$/ $p = 0.044$, Table 1).

Watching videos using HMD-type VR devices inevitably causes visual discomfort [21,22]. First, discomfort such as simulator sickness, headache, and gastrointestinal symptoms are caused by the visual-vestibular conflict. Second, mechanical myopia induced by the environment inside VR devices can cause accommodative spasms, and this can lead to secondary myopia, fuzzy images, and difficulty focusing. Third, a mismatch between the viewer's visual system and VR device can arise. In particular, when the difference between the viewer's interpupillary distance and the distance to the optical center of the VR device lens is large, blur and diplopia can arise. Even if the difference is small, it can still cause eye fatigue, even though the visual acuity does not change. This condition can lead to changes in phoria and fixation disparity. Anoh-Tanon et al. [23] found that people with binocular vision problems or poor eye alignment have reduced fixation ability, which can impair body stability. Kim et al. [24] argued that changes in phoria can cause neck-muscle tension, which in turn leads to changes in the neck position and requires recalibration of the vestibular signals to adjust the posture. As such, the visual side effects arising from a VR environment are evident and can be a major cause of temporary visual problems that can impair body stability. Because HMD-type VR images require focusing on the 3D VR implemented through the VR device, the position of the rendered image depends on the lens power built into the VR device. Therefore, in order to minimize visual fatigue and cyber sickness when watching VR images, it is important to set the distance to the optical center of the VR headset lens to the most comfortable position for each individual before they begin watching.

In this study, the conditions for refractive error correction as well as the distance to the optical centers of VR-device lenses were defined. The results of the study showed that body

stability became aggravated under the conditions under which the distance to the optical centers of VR-device lenses caused the most discomfort and those under which hyperopia was induced. Specifically, the sway range in the anterior-posterior direction increased significantly only under the DOCD-2D condition (the conditions under which the distance to the optical center caused discomfort and induced hyperopia) ($p < 0.05$, Figure 4A). A recent study investigated the relationship between hyperopia and body stability and reported the following interesting result: mild hyperopia induced with $-1.00$ D spherical lenses reduced body stability in spite of an average visual acuity of 1.0 or more [25]. The authors explained that the continuous eye adjustment involved in refocusing caused an imbalance in the autonomic nervous system, which affected the vestibular system involved in posture control. Their results confirmed that VR viewers' hyperopic refractive error is a type of refractive error that interferes with body stability.

Edwards [26] reported that body instability increased by more than 50% when myopic blur was induced in 50 participants in an experiment in which they used a +5.00 D spherical lens. Similarly, Paulus et al. [27] also reported that body instability increased by about 25% when myopia was induced using +4.00 D and +6.00 spherical lenses compared to instability before induction of myopia. However, no such significant difference was found in our study under myopia-inducing conditions, although blurred visual information caused by myopia is a visual factor that interferes with the body stability. Apparently, this result arises from offsetting the mechanical myopia induced by the environment within the VR device. Based on the result, additional research to determine the proper diopteric power of the optical lens by considering the viewers' accommodative function offsetting effect is needed when developing an HMD-type VR device.

This study has the following limitations. First, the induced refractive error was limited to only $\pm 2.00$ D. There is a need to investigate not only the impact of various levels of myopia and hyperopia, but also the effects of astigmatism and anisometropia. Second, the analysis why only the distance positive fusional vergence was significant in the analysis of the correlation between the visual function and body stability is limited. Third, the real-time monitoring of changes in eye function while wearing the VR headset was negatively affected by technical limitations. Future studies are planned to address the limitations of this study.

## 5. Conclusions

The present study investigated the postural-control ability of 31 adults using a BTrackS device under various optical conditions, including comfortable and uncomfortable optical-center distances of VR-device lenses, as well as different refractive error statuses. Based on the our findings, it is evident that the distance between the optical centers of VR-device lenses and the refractive error status of users significantly influence body stability (sway-path length, range anterior posterior of sway, and velocity maximum of sway) during VR viewing. Therefore, allowing users to maintain body stability during VR viewing by setting the optimal optical center distance to the VR headset lens for each individual viewer is important. Particularly, viewers with hyperopic refraction error should be careful because the negative effect on body stability could be worsened. According to the sensory-conflict theory, cyber sickness arises as a result of conflicts between visual and vestibular cues because the experience perceived in the virtual environment is more artificial than that in the real world [3]. Although this factor was not analyzed in this study, our results suggest the possibility that uncomfortable optical center-distance settings and residual refractive errors in VR users may exacerbate conflicts between visual and vestibular cues, potentially leading to increased cyber sickness and negative impacts on body stability. As of recently, new HMD-VR technologies offer greatly improved visual and tracking resolutions; however, users may still suffer from cyber sickness and body instability, likely caused by a combination of factors including VR hardware (visual field, display resolution, latency, motion tracking) [28–30], VR content (optical flow, graphic realism, reference frame, and task) [30,31], and user characteristics (age, gender, prior VR experiences and motion

sickness susceptibility) [31]. Additionally, our results suggest that the refractive-error status of the user and the optical center distance between the lenses in the VR device should also be considered as important factors contributing to increased body instability during VR viewing. These findings are expected to be important considerations for developers of VR head-mounted devices. Therefore, in our next research effort, we intend to investigate the correlation between cyber sickness and body stability during the viewing of VR videos, which could be a crucial step in establishing human-friendly 3D virtual environments.

**Author Contributions:** Conceptualization, H.-G.K. and S.-Y.K.; methodology, G.-S.J. and S.-Y.K.; validation, G.-S.J., H.-G.K. and S.-Y.K.; investigation, G.-S.J. and S.-Y.K.; resources, H.-G.K. and S.-Y.K.; data curation, G.-S.J., and S.-Y.K.; writing—original draft preparation, G.-S.J.; writing—review and editing, H.-G.K. and S.-Y.K.; visualization, H.-G.K. and S.-Y.K.; supervision, S.-Y.K. All authors have read and agreed to the published version of the manuscript.

**Funding:** This research was supported by the "Regional Innovation Strategy (RIS)" through the National Research Foundation of Korea (NRF) funded by the Ministry of Education (MOE) (2022RIS-005).

**Data Availability Statement:** The data presented in this study are available upon request from the corresponding author.

**Acknowledgments:** We appreciate the financial support by the "Regional Innovation Strategy (RIS)" through the National Research Foundation of Korea (NRF) funded by the Ministry of Education (MOE).

**Conflicts of Interest:** The authors declare no conflicts of interest.

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
