# Peer review of "Dependence of Body Stability on Optical Conditions during VR Viewing"

_electronics, doi:10.3390/electronics13101812_

Round 1

Reviewer 1 Report

Comments and Suggestions for Authors

It is an interesting paper that is worth publishing but it does require some more work to be fully ready for publication

I have listed a series of comments below that should be addressed to make the paper better for the readership

- figure 2, please correct the typo in the x axis (optcal) and Y axis should start at 0, the graph should also mention what error bars represent (ex. SD, or 95% CI)

- figure 3 A, Y axis start at zero, and B should have the same Y axis as A in scale so one can see that ML sway is lower than AP, please address the error bar as above

- line 288, this is not a collision but a conflict, see work of T.A. Stoffregen

- despite being addressed in the introduction, and in the conclusion, there is no data collected on cybersickness and thus, the abstract should not state that the experiment minimized it. The results only show results for body stability.

- This is for this reason that I consider the major to require a major revision because a great emphasis is placed on the reduction of cybersickness without actually documenting it among the 31 participants.

Author Response

Response to Reviewer 1 Comments

Comments and Suggestions for Authors : 

It is an interesting paper that is worth publishing but it does require some more work to be fully ready for publication. I have listed a series of comments below that should be addressed to make the paper better for the readership

Response : Thank you for reviewing our manuscript electronics-2978451, titled “Dependence of Body Stability on Optical Conditions during VR Viewing,” submitted for publication in Electronics. Thank you for postive eval to the authors' papers and for the detailed review.

Point 1: figure 2, please correct the typo in the x axis (optcal) and Y axis should start at 0, the graph should also mention what error bars represent (ex. SD, or 95% CI)

Response for point 1 : Based on the reviewer’s advice, we have revised scale of Y axis in Figure 2 (start at 0), and we have corrected typographical errors (optcal --> optical). Note that Figure 2 has been changed to Figure 3. Also, we added a footnote in Figures that the error bars represent the standard deviation, as follows; (Please check lines 226 and 228-230 in tracked changes manuscript)

“Error bars were expresses as mean ± SD”

Point 2: figure 3 A, Y axis start at zero, and B should have the same Y axis as A in scale so one can see that ML sway is lower than AP, please address the error bar as above

Response for point 2 : We have revised the y-axis scale of both Figure 3A and B to start at 0. Also, we added a footnote in Figures that the error bars represent the standard deviation. Note that Figure 3 has been changed to Figure 4. (Please check lines 250 and 252-254)

Point 3: line 288, this is not a collision but a conflict, see work of T.A. Stoffregen

Response for point 3: Based on the reviewer’s advice, we have revised “collision” to “conflict.”  (Please check line 319)

Point 4: despite being addressed in the introduction, and in the conclusion, there is no data collected on cybersickness and thus, the abstract should not state that the experiment minimized it. The results only show results for body stability.

Response for point 4: We totally agree with the reviewer's opinion. We have deemed it appropriate to delete the mentioned content about cyber sickness, so we have removed the relevant sentence from the abstract and discussion sections. 

Deleted content: “and cyber sickness” (Please check line 26), “worsens cyber sickness symptoms and” (Please check line 381)

Point 5: This is for this reason that I consider the major to require a major revision because a great emphasis is placed on the reduction of cybersickness without actually documenting it among the 31 participants.

Response for point 5: During VR viewing, while there is a possibility that cyber sickness may have affected the postural instability examined in this study, I agree that there is a limitation in not providing direct data. We have added limitations to the conclusion section regarding these points, and further proposed topics for future research, as follows; (Please check lines 413-419)

“Although not analyzed in this study, our results suggest the possibility that cyber sickness, which may be exacerbated when viewing VR under optically inappropriate visual conditions, could have a negative impact on body stability. Therefore, in our next research, we intend to investigate the correlation between cyber sickness and body stability during the viewing of VR videos, which could be a crucial step in establishing human-friendly 3D virtual environments.”

We thank you again for your attention to the authors' papers and for the detailed review. We’ve done our best to provide reviewers with a reasonable answer. 

< The End Reply>

Reviewer 2 Report

Comments and Suggestions for Authors

The study "Dependence of Body Stability on Optical Conditions during VR Viewing" explores the impact of different optical conditions related to Virtual Reality (VR) headsets on body stability. The manuscript is well written and well organized, except for the abstract and conclusion.

1. The abbreviations in the abstract make it hard to read and confusing.

2. The conclusion is incomplete and can be improved. The conclusion is short and does not provide much information; as mentioned earlier, the conclusion needs to be elaborated. The conclusion could be improved by providing a more concise and direct summary of the main findings and their implications. Additionally, it would be beneficial to include a statement about the limitations of the study and suggestions for future research.

3. Some insight into future directions can also be beneficial.

4.Increasing the sample size and considering a broader optical spectrum would also be beneficial.

Author Response

Response to Reviewer 2 Comments

Comments and Suggestions for Authors : 

The study "Dependence of Body Stability on Optical Conditions during VR Viewing" explores the impact of different optical conditions related to Virtual Reality (VR) headsets on body stability. The manuscript is well written and well organized, except for the abstract and conclusion.

Response : Thank you for reviewing our manuscript electronics-2978451, titled “Dependence of Body Stability on Optical Conditions during VR Viewing,” submitted for publication in Electronics. Thank you for your attention to the authors' papers and for the detailed review.

Point 1: The abbreviations in the abstract make it hard to read and confusing.

Response for point 1 : I agree that the abundance of abbreviations may cause some confusion. However, the abbreviations are appropriately expressed, and we ask for your understanding that this is inevitable due to the word numbers limit in the abstract. To reduce confusion, we have rewritten the abbreviations in parentheses as full terms. (Please check lines 16-19 in tracked changes manuscript)  

Point 2: The conclusion is incomplete and can be improved. The conclusion is short and does not provide much information; as mentioned earlier, the conclusion needs to be elaborated. The conclusion could be improved by providing a more concise and direct summary of the main findings and their implications. Additionally, it would be beneficial to include a statement about the limitations of the study and suggestions for future research.

Response for point 2 : In accordance with the reviewer's advice, I have revised the conclusion section to include the main findings, limitations, and future research topics, as follows; 

“The present study investigated the postural control ability of 31 adults using a BTracks device under various optical conditions, including comforted and discomforted optical center distances of VR device lenses, as well as different refractive error statuses. Based on the our findings, it is evident that the distance between the optical centers of VR device lenses and the refractive error status of users significantly influence body stability (sway path length, range anterior posterior of sway, and velocity maximum of sway) during VR viewing.”(Please check lines 403-409) 

Although not analyzed in this study, our results suggest the possibility that cyber sickness, which may be exacerbated when viewing VR under optically inappropriate visual conditions, could have a negative impact on body stability. Therefore, in our next research, we intend to investigate the correlation between cyber sickness and body stability during the viewing of VR videos, which could be a crucial step in establishing human-friendly 3D virtual environments.” (Please check lines 413-419)

Point 3: Some insight into future directions can also be beneficial.

Response for point 3: In the future research, we believe that analyzing the correlation between cyber sickness and postural instability is an important topic. We have added this point to the conclusion section. (Please check lines 413-419) 

Point 4: Increasing the sample size and considering a broader optical spectrum would also be beneficial.

Response for point 4: We completely agree with the reviewer's opinion. We hope you understand that conducting additional experiments is not feasible at present. However, we will endeavor to recruit an adequate number of participants for future research. Additionally, we have included the following limitation regarding a broader optical spectrum in the discussion section, as follows; (Please check lines 395-396) 

“There is a need to investigate not only the impact of various levels of myopia and hyperopia but also the effects of astigmatism and anisometropia.”  

 We thank you again for your attention to the authors' papers and for the detailed review. We’ve done our best to provide reviewers with a reasonable answer. 

< The End Reply>

Reviewer 3 Report

Comments and Suggestions for Authors

This work present an opporunity to improve the uses of VR headset , althoug could be improve  :

- At section 2, at the beginning, include a visual description of the  method used, for example: use a flowchart

- In section 2.6 Could you present a data chart , from SPSS results.

- At conclusions section, it can be improved if you describe more about how or why "these findings are expected to be important considerations for developers of VR head-mounted devices."

Comments on the Quality of English Language

Minor review at introduction section

Author Response

Response to Reviewer 3 Comments

Comments and Suggestions for Authors : 

This work present an opporunity to improve the uses of VR headset , althoug could be improve :

Response : Thank you for reviewing our manuscript electronics-2978451, titled “Dependence of Body Stability on Optical Conditions during VR Viewing,” submitted for publication in Electronics. Thank you for your attention to the authors' papers and for the detailed review.

Point 1: At section 2, at the beginning, include a visual description of the  method used, for example: use a flowchart

Response for point 1 : In accordance with the reviewer's advice, we have added the flowchart, as follows; (Please check lines 96-97 in tracked changes manuscript)

Figure 1. The flow chart of the present study.

Point 2: In section 2.6 Could you present a data chart , from SPSS results.

Response for point 2 : We have added detailed results of the SPSS analysis, including F-values and post-hoc analysis results in Results section, as follows;

“The sway path length increased significantly for COCD-2D, DOCD, DOCD+2D, and DOCD-2D compared to that of COCD (repeated-measures ANOVA, F = 3.887/p = 0.009). A post-hoc analysis revealed that p = 0.012 for COCD vs. COCD-2D, p = 0.001 for COCD vs. DOCD, p = 0.015 for COCD vs. DOCD+2D, and p=0.001 for COCD vs. DOCD-2D condition. However, compared to the COCD+2 condition, the sway path length tended to increase, but there was no statistically significant difference (p = 0.258 for COCD vs. COCD+2D).” (Please check lines 217-223)  

“The variations in the range of anterior-posterior sway were significantly different only under the DOCD-2D condition compared to the COCD condition (repeated-measures ANOVA, F = 2.732/p = 0.032, Figure 4A). A post-hoc analysis revealed that p = 0.019 for COCD vs. COCD-2D. The variations in the range of medial-lateral sway were statistically insignificant (repeated-measures ANOVA, F = 0.718/p = 0.616, Figure 4B).” (Please check lines 241-246)  

“The differences in the velocity maximum of sway were statistically significant under the conditions of DOCD, DOCD+2D, and DOCD-D compared to that of COCD condition (repeated-measures ANOVA, F = 4.030/p = 0.008). A post-hoc analysis revealed that p = 0.001 for COCD vs. DOCD, p = 0.007 for COCD vs. DOCD+2D, and p = 0.005 for COCD vs. DOCD-2D condition.” (Please check lines 266-271)  

Point 3: At conclusions section, it can be improved if you describe more about how or why "these findings are expected to be important considerations for developers of VR head-mounted devices."

Response for point 3: In accordance with the reviewer's advice, I have revised the conclusion section to include the main findings, limitations, and future research topics, as follows; 

“The present study investigated the postural control ability of 31 adults using a BTracks device under various optical conditions, including comforted and discomforted optical center distances of VR device lenses, as well as different refractive error statuses. Based on the our findings, it is evident that the distance between the optical centers of VR device lenses and the refractive error status of users significantly influence body stability (sway path length, range anterior posterior of sway, and velocity maximum of sway) during VR viewing.”(Please check lines 403-409) 

Although not analyzed in this study, our results suggest the possibility that cyber sickness, which may be exacerbated when viewing VR under optically inappropriate visual conditions, could have a negative impact on body stability. Therefore, in our next research, we intend to investigate the correlation between cyber sickness and body stability during the viewing of VR videos, which could be a crucial step in establishing human-friendly 3D virtual environments.”(Please check lines 413-419)

We thank you again for your attention to the authors' papers and for the detailed review. We’ve done our best to provide reviewers with a reasonable answer. 

< The End Reply>

Reviewer 4 Report

Comments and Suggestions for Authors

First of all, I would like to congratulate the author for the work. The research is solid and very interesting. The only contribution I see that is required is a more extended description of the roller coaster video projected in the glasses, in order to analyze the experiment. I also see a lack of concluding material where I observe that the results section is absorbing this conclusions section. Therefore, I suggest to the authors a better explanation of the experiment and a reformulation of the results and conclusions sections.

Author Response

Response to Reviewer 4 Comments

Comments and Suggestions for Authors : 

First of all, I would like to congratulate the author for the work. The research is solid and very interesting. The only contribution I see that is required is a more extended description of the roller coaster video projected in the glasses, in order to analyze the experiment. I also see a lack of concluding material where I observe that the results section is absorbing this conclusions section. Therefore, I suggest to the authors a better explanation of the experiment and a reformulation of the results and conclusions sections.

Response :  Thank you for reviewing our manuscript electronics-2978451, titled “Dependence of Body Stability on Optical Conditions during VR Viewing,” submitted for publication in Electronics. Thank you for your attention to the authors' papers and for the detailed review.

Based on the reviewer’s advice, we have added a description of the roller coaster video used in this study, as follows; (Please check lines 199-203 in tracked changes manuscript) 

“The roller coasters video used this study was designed for 360-degree virtual reality (VR) in 4K resolution, 60 frames per second (fps), and 3D format which was presented to both eyes simultaneously. During the posture assessment, the same 30-second segment from the entire roller coaster video was edited and provided for viewing to all participants.” 

Also, We have rewritten the Results and Conclusion sections as follows; 

“The sway path length increased significantly for COCD-2D, DOCD, DOCD+2D, and DOCD-2D compared to that of COCD (repeated-measures ANOVA, F = 3.887/p = 0.009). A post-hoc analysis revealed that p = 0.012 for COCD vs. COCD-2D, p = 0.001 for COCD vs. DOCD, p = 0.015 for COCD vs. DOCD+2D, and p=0.001 for COCD vs. DOCD-2D condition. However, compared to the COCD+2 condition, the sway path length tended to increase, but there was no statistically significant difference (p = 0.258 for COCD vs. COCD+2D).” (Please check lines 217-223)  

“The variations in the range of anterior-posterior sway were significantly different only under the DOCD-2D condition compared to the COCD condition (repeated-measures ANOVA, F = 2.732/p = 0.032, Figure 4A). A post-hoc analysis revealed that p = 0.019 for COCD vs. COCD-2D. The variations in the range of medial-lateral sway were statistically insignificant (repeated-measures ANOVA, F = 0.718/p = 0.616, Figure 4B).” (Please check lines 241-246)  

“The differences in the velocity maximum of sway were statistically significant under the conditions of DOCD, DOCD+2D, and DOCD-2D compared to that of COCD condition (repeated-measures ANOVA, F = 4.030/p = 0.008). A post-hoc analysis revealed that p = 0.001 for COCD vs. DOCD, p = 0.007 for COCD vs. DOCD+2D, and p = 0.005 for COCD vs. DOCD-2D condition.” (Please check lines 266-271) 

“The present study investigated the postural control ability of 31 adults using a BTracks device under various optical conditions, including comforted and discomforted optical center distances of VR device lenses, as well as different refractive error statuses. Based on the our findings, it is evident that the distance between the optical centers of VR device lenses and the refractive error status of users significantly influence body stability (sway path length, range anterior posterior of sway, and velocity maximum of sway) during VR viewing.”(Please check lines 403-409) 

Although not analyzed in this study, our results suggest the possibility that cyber sickness, which may be exacerbated when viewing VR under optically inappropriate visual conditions, could have a negative impact on body stability. Therefore, in our next research, we intend to investigate the correlation between cyber sickness and body stability during the viewing of VR videos, which could be a crucial step in establishing human-friendly 3D virtual environments.”(Please check lines 413-419)

We thank you again for your attention to the authors' papers and for the detailed review. We’ve done our best to provide reviewers with a reasonable answer. 

< The End Reply>

Round 2

Reviewer 1 Report

Comments and Suggestions for Authors

Line 413, it is not cybersickness detrimental to body stability but the visual conflict. Literature on the topic of body posture control and vision should be revisited.

Unfortunately, due to this misalignment with the literature, the paper is not yet ready for publication. Authors can seek in Gait and Posture journal.

Reviewer 2 Report

Comments and Suggestions for Authors

The authors have addressed all the concerns I have raised. I appreciate your work.

Author Response

We thank you again for your attention to the authors' papers and for the detailed review. 

Round 3

Reviewer 1 Report

Comments and Suggestions for Authors

thanks for making the appropriate modifications to your manuscript. Despite being a phenomenological study, the point that was raised was on a misalignment between the results obtained and a possible cause-and-effect relation, which was not the case. This is why the modifications were requested.

There might still be typos to be addressed but I'm the person to suggest those.